# Role of the EUS in the Treatment of Biliopancreatic Disease in Patients with Surgically Altered Anatomy

**DOI:** 10.3390/diagnostics15212707

**Published:** 2025-10-26

**Authors:** Marcello Cintolo, Edoardo Forti, Giulia Bonato, Michele Puricelli, Lorenzo Dioscoridi, Marianna Bravo, Camilla Gallo, Francesco Pugliese, Andrea Palermo, Alessia La Mantia, Massimiliano Mutignani

**Affiliations:** Endoscopy Unit, ASST Grande Ospedale Metropolitano Niguarda, 20162 Milan, Italy; giulia.bonato@ospedaleniguarda.it (G.B.); lorenzo.dioscoridi@ospedaleniguarda.it (L.D.); marianna.bravo@ospedaleniguarda.it (M.B.); camilla.gallo@ospedaleniguarda.it (C.G.); francesco.pugliese@ospedaleniguarda.it (F.P.); andrea.palermo@ospedaleniguarda.it (A.P.); alessia.lamantia@ospedaleniguarda.it (A.L.M.); massimiliano.mutignani@ospedaleniguarda.it (M.M.)

**Keywords:** EUS, surgically altered anatomy, EDGE

## Abstract

Background: The rising prevalence of gastric, biliary, and pancreatic surgeries has led to an increasing population of patients with surgically altered anatomy (SAA). In this setting, conventional endoscopic retrograde cholangiopancreatography (ERCP) is often limited by anatomical barriers, resulting in high rates of technical failure and complications. While device-assisted enteroscopy (DAE) has expanded therapeutic possibilities, its efficacy remains modest in complex reconstructions. Methods: This review analyzed recent literature from PubMed, Embase, and Scopus up to April 2025, focusing on diagnostic and therapeutic roles of endoscopic ultrasound (EUS) in SAA. Particular attention was given to cases where standard endoscopic, percutaneous, or surgical techniques failed and to studies comparing EUS-guided approaches with alternative modalities. Results: EUS has transitioned from a primarily diagnostic modality to a versatile therapeutic platform in SAA. Techniques such as EUS-guided rendezvous, antegrade drainage, and hepaticogastrostomy have shown technical and clinical success rates exceeding 80–90%, often comparable or superior to interventional radiology, while reducing the need for external drains. Innovative procedures, including EUS-directed transgastric ERCP (EDGE) and EUS-directed enteroenteric bypass (EDEE), have transformed the management of Roux-en-Y gastric bypass and bilioenteric anastomoses, providing durable and reusable access for repeated interventions. Despite these advances, EUS-guided interventions remain technically demanding, requiring advanced endoscopic and radiologic skills, specialized devices, and are best performed in tertiary referral centers. Conclusions: EUS has redefined the treatment paradigm of biliopancreatic diseases in patients with SAA, increasingly emerging as the preferred minimally invasive approach when conventional techniques fail. Future developments will focus on dedicated devices, standardized guidelines, and structured training programs to optimize outcomes. Multidisciplinary collaboration and centralization in high-volume centers remain essential to ensure safety, efficacy, and reproducibility.

## 1. Introduction

The increasing use of surgical interventions has resulted in a growing number of patients with surgically altered anatomy (SAA) presenting for endoscopic evaluation and therapy. SAA refers to situations in which the continuity between the stomach, or its remnant, and the duodenum is disrupted as a consequence of surgical intervention [1]. Common examples include various Roux-en-Y reconstructions (such as partial or total gastrectomy, gastric bypass, hepaticojejunostomy, Roux-en-Y liver transplantation, and selected cases of liver resection, particularly left hepatectomy), Billroth II gastrectomy, and Whipple pancreaticoduodenectomy.

SAA presents significant challenges to the performance of both diagnostic and therapeutic pancreato-biliary endoscopy [2]. Endoscopic ultrasound (EUS) and EUS-tissue acquisition (EUS-TA) may be limited in the diagnostic setting by the inability to closely access the pancreatic head or the common bile duct [3,4]. Conversely, in the therapeutic field, EUS has become a central modality, changing the treatment paradigm of biliary and pancreatic disorders in patients with SAA.

Endoscopists should acquire a comprehensive understanding of surgical modifications and their implications for both diagnostic and therapeutic endoscopic procedures.

This review aims to summarize the current evidence on the limitations and performance of diagnostic EUS in patients with SAA and to explore the role of interventional EUS when retrograde access to the biliary or pancreatic ducts is not feasible due to SAA.

## 2. Materials and Methods

A systematic search of PubMed, Embase and SCOPUS databases was performed. To enhance transparency and reproducibility, we combined keywords and Medical Subject Headings (MeSH) using Boolean operators. The search was performed using the following string “endoscopic ultrasound” (OR “endoscopic ultrasonography” OR “EUS” OR “Endoscopic Retrograde Cholangiopancreatography” OR “ERCP” OR “Endoscopic ultrasound-directed transgastric ERCP (EDGE)” OR “EDGE” OR “Endoscopic Ultrasound-Directed Transenteric ERCP (EDEE)” OR “EDEE” OR “Fine Needle Aspiration” OR “Fine Needle Aspiration (FNA)” OR “FNA” OR “Fine Needle Biopsy (FNB)” OR “FNB” OR “lumen apposing metal stents” OR “LAMS”) AND “altered anatomy” (OR “post-surgical anatomy” OR “surgically altered anatomy” OR “gastric bypass” OR “bariatric surgery” OR “Billroth II” OR “Roux-en-Y” OR “surgical anastomosis” OR “pancreatectomy”). The review was conducted in accordance with the PRISMA 2020 statement and, given the exploratory scope, followed the PRISMA-ScR checklist for scoping reviews where applicable. Only English-language, full-text publications were considered eligible for this review. Both prospective and retrospective cohort studies as well as randomized controlled trials, case series or reports were included. International guidelines and systematic reviews or meta-analyses were also considered.

## 3. Postsurgical Anatomy

The first step toward performing a high-quality and appropriate procedure is having a solid knowledge of the different SAAs and their implications. From a biliopancreatic perspective, patients with SAA can be roughly divided into two categories: patients with a native papilla (Billroth I and II, gastric bypass, gastrectomy with Roux-en-Y reconstruction) and patients with a biliodigestive anastomosis (pancreaticoduodenectomy, hepaticojejunostomy). Another way to categorize these patients is based on the accessibility of the papillary region/anastomosis: patients with easily accessible papilla/anastomosis (BI and BII, DCP) and patients with not easily accessible papilla/anastomosis (gastric bypass, gastrectomy with Roux-en-Y reconstruction, hepaticojejunostomy).

### 3.1. Billroth II (BII) Gastrectomy

BII gastrectomy has historically been the most common reconstruction for complicated peptic ulcer disease and still plays a role in gastric cancer surgery; however, Roux-en-Y gastrectomy is now commonly preferred in this setting.

This intervention is characterized by a termino-lateral gastrojejunostomy, leading to the creation of an afferent and an efferent loop. The entrance of the afferent loop can vary in position, depending on how the reconstruction is performed: if the loop is anastomosed in an isoperistaltic direction, the entry is located near the lesser curve (Figure 1). In the isoperistaltic type, the entry site is located near the greater curvature [5].

The length of the afferent loop is also important, since a long or angulated afferent limb not only makes endoscopic intubation more difficult but also predisposes to stasis and the so-called “afferent loop syndrome.” To address these complications and prevent biliary reflux, a variant of Braun’s enteroenterostomy can be considered.

### 3.2. Gastrectomy (Total or Distal) with Roux-en-Y Reconstruction (RYG)

RYG is the paramount surgical procedure in managing gastric cancer, with the aim of removing tumors while preserving healthy tissue and function completely. Surgical variants include total, proximal, distal, and pylorus-preserving distal gastrectomy; the choice is made according to tumor location, extent, and patient factors [6].

Total gastrectomy is considered for tumors involving a significant portion of the proximal stomach, especially those extending along most of the lesser or greater curvature; distal gastrectomy is the best option for cancers located in the antrum or in the distal part of the body. The reconstruction consists of a gastrojejunal anastomosis (or esophageal-jejunal anastomosis in case of total gastrectomy) with an alimentary loop (the Roux loop), and the creation of a jejunojejunal anastomosis placed at 40–60 cm below, with the biliary loop (the Y-loop) that receives bile and pancreatic fluids (Figure 2) [7].

### 3.3. Roux-en-Y Gastric Bypass (RYGB)

RYGB is the most widely performed bariatric procedure worldwide. It consists of creating a small gastric pouch anastomosed to the jejunum, bypassing the excluded stomach and duodenum (Figure 3). This operation was originally designed for weight loss but is now equally valued for its metabolic benefits in patients with type 2 diabetes and severe obesity. Variations in limb length are frequent: long biliopancreatic or alimentary limbs are often employed in super-obese patients or those with severe metabolic disease to increase malabsorptive effects. An alternative is the mini-gastric bypass, in which a long gastric pouch is anastomosed directly to a jejunal loop. Despite its effectiveness in reducing calorie intake and weight loss, it carries the risk of bile reflux, which remains a subject of ongoing debate and study [8].

For patients affected by extreme obesity with refractory metabolic disease, biliopancreatic diversion (BPD) and its modern variant, as the single-anastomosis duodeno-ileal bypass with sleeve gastrectomy (SADI-S), can be considered. Despite its effectiveness in reducing calorie intake and causing weight loss, it can carry significant risks of protein-calorie malnutrition and micronutrient deficiencies, necessitating lifelong follow-up and supplementation. BPD, for the reasons mentioned above, is currently rarely employed; in addition, in some cases, the resulting malnutrition may be associated with significant morbidity and, in rare instances, can even be fatal [9].

### 3.4. Pancreaticoduodenectomy

Pancreaticoduodenectomy is the gold standard for treating tumors of the head of the pancreas, ampullary lesions, distal bile duct tumors, and selected duodenal lesions. This complex surgery entails en bloc resection of the pancreatic head, duodenum, gallbladder, distal bile duct, and frequently part of the stomach, followed by reconstruction of gastrointestinal continuity with pancreaticojejunostomy, hepaticojejunostomy, and gastrojejunostomy (Figure 4) [10].

A well-established variant is the pylorus-preserving pancreaticoduodenectomy (PPPD), which spares the gastric antrum and pyloric channel, reconnecting via a duodenojejunostomy. While PPPD maintains more physiological gastric emptying, evidence on delayed gastric emptying (DGE) is mixed: some series suggest a higher incidence compared to the classic Whipple, though overall morbidity and oncologic outcomes are similar. Randomized data confirm comparable survival and complication rates, though the choice between techniques should be individualized, considering tumor location and patient profile.

### 3.5. Hepaticojejunostomy (HJ)

HJ is the gold standard for biliary reconstruction in a variety of settings, such as bile duct injury, cholangiocarcinoma resection, liver transplantation, and treatment of benign biliary strictures. The procedure involves a Roux-en-Y anastomosis between hepatic bile ducts and a jejunal limb, establishing durable biliary drainage through meticulous mucosa-to-mucosa technique (Figure 5) [11].

In surgical reconstructions, the anastomosis can be created to one, two, or even three bile ducts, according to the involvement of the biliary tree and the extent of hepatic resection. A monoduct anastomosis is considered for lesions involving the common bile duct below the hilar confluence, while biduct or triduct (rarely quadriduct) anastomoses are required in cases of a separation between right and left ducts, or between the other segmental ducts. Although these techniques provide adequate bile drainage, they inevitably result in multiple small anastomotic orifices. From the endoscopist’s perspective, this substantially increases technical complexity, as each duct must be managed separately: cannulation, dilation, and stenting of each individual anastomosis become challenging and often less effective compared to single-duct reconstructions [12].

Alternate reconstructive strategies, including end-to-end biliary repair or T-tube placement, may serve select strictures or injuries; however, when ductal injury exceeds critical thresholds (e.g., long transections close to bifurcation), Roux-en-Y HJ remains preferred [13].

## 4. Diagnostic EUS and EUS-TA in Altered Anatomy

Data on the safety and feasibility of EUS in patients with altered anatomy are still limited. Major limitations can be encountered in the study of the pancreatic head or the extrahepatic biliary tract due to the challenges of closely approaching these structures with the transducer. Pancreatic body-to-tail lesions are more easily detected and punctured [14].

In patients with distal gastrectomy, pancreatoduodenectomy, or gastrojejunostomy, initial attempts to detect the target lesion should be made through the residual stomach, even when the lesion is located around the pancreatic head. If this approach is unsuccessful, scanning via the jejunal limb can be subsequently attempted. In cases of total gastrectomy, scanning can begin at the esophago-jejunal anastomosis, with the echoendoscope carefully advanced into the jejunal limb to localize the target lesion [14].

A retrospective study on EUS, both with radial or linear probes, reported limitations in 10/39 patients with Billroth II anatomy in evaluating the common bile duct (CBD) or the head of pancreas (HOP) because intubation of the afferent limb failed, or was not attempted. In 13/18 patients with Roux-en-Y anatomy, the proximal duodenum was not reached, and the HOP and CBD were not imaged. For 6/7 patients, Roux-en-Y gastric bypass, the HOP and CBD were not viewed [3].

In a retrospective series on EUS on altered anatomy, visualization of the head of the pancreas was significantly impacted by total gastrectomy, Billroth II, and Roux-en-Y with success rates of 6.7%, 53.7%, and 57.1%, respectively [4].

As regards fine-needle tissue acquisition with aspiration or biopsy (FNA/FNB), overall technical success and diagnostic accuracy of EUS-TA were 78.2% and 71.3%, respectively [4]. Reasons for EUS-TA technical failure were mainly failed visualization, followed by insufficient proximity between the probe and the lesion. Pancreatic body-to-tail lesions are instead more easily detected and punctured. Examination of the pancreatic body and tail was impaired in esophagectomy and total gastrectomy, 82.4% and 71.4%, respectively [4].

Some authors reported acceptable results both in lesion visualization and tissue acquisition with forward-viewing linear-array echoendoscopes [15,16]; although such devices are rarely available at the Endoscopy Service. Moreover, performances in cases of Roux-en-Y gastric surgery remain low [16]. In very selected cases, when both transcutaneous CT-guided biopsy or EUS alone are unsuccessful due to the altered anatomy, a modified EDGE can be considered in order to allow for EUS-FNA of a pancreatic mass [17].

## 5. Management of Biliary Diseases in Patients with SAA

Indications for biliary interventions in this population are diverse. The occurrence of bile duct stones has been reported in approximately 0.75% of patients following Roux-en-Y reconstruction gastrectomy [18] and 1.4–2% after Roux-en-Y gastric bypass [19,20]. Hepaticojejunostomy anastomotic stricture (HJAS) is a long-term complication after HJ that is estimated to occur in approximately 12.5% of patients within two years of surgery [21].

ERCP in patients with SAA is well recognized as a technical challenge, with higher failure and complication rates compared with native anatomy. Large case series have reported technical failure rates of 10–20% in Billroth II anatomy [22,23], 18% in Whipple anatomy [24], and 12–25% in Roux-en-Y reconstructions [22,25].

In this context, EUS offers unique advantages, enabling direct transmural access to the bile or pancreatic duct, or to the excluded stomach or jejunal loop, thus allowing subsequent ERCP-based interventions [26].

### 5.1. Interventional Radiology

Interventional radiology has long been the cornerstone of biliary decompression in patients where endoscopic access is not feasible. In SAA, percutaneous approaches continue to represent an important therapeutic option, either as a bridge to definitive therapy or as a long-term solution when endoscopic expertise is unavailable.

#### 5.1.1. Percutaneous Transhepatic Cholangiography (PTC)

PTC is the diagnostic and therapeutic entry point in this domain. Using fluoroscopic guidance, intrahepatic ducts are punctured, opacified, and accessed with guidewires and catheters. Several series have reported high feasibility rates for PTC, often exceeding 90%. However, comparative studies indicate that its technical success does not appear to be superior to that of EUS-biliary drainage (EUS-BD), with both techniques showing comparable reliability. Conversely, EUS-BD has been associated with lower reintervention rates and greater cost-effectiveness [27].

#### 5.1.2. Percutaneous Transhepatic Biliary Drainage (PTBD)

PTBD extends PTC by placing external or internal–external catheters to drain the biliary system. It achieves decompression reliably in >90% of cases with dilated ducts and in 70–80% of those with nondilated ducts. Adverse events (AEs), however, are not negligible: reported rates range from 10% to 30%, with major complications including hemobilia, bile peritonitis, sepsis, and pneumothorax. Mortality directly related to the procedure is rare (<1%), but morbidity is significant [28].

Cantwell and colleagues published a thirty-year retrospective study in 2008 about their experience with percutaneous balloon dilation (PBBD) in patients with benign postoperative biliary strictures. This long-term study confirms that PBBD is technically feasible and generally safe, with a low rate of major complications (2%). Nevertheless, the long-term probability of avoiding restenosis declines gradually over time, and almost half of patients develop restenosis at 5-year follow-up [29].

#### 5.1.3. Percutaneous Stenting

Percutaneous stent placement can be performed following initial drainage, allowing internalization of bile flow and improving patient comfort compared with external drains. Technical success rates for stenting are usually above 90%, while clinical success is reported in 80–85% of cases. Nevertheless, reintervention rates remain high, as stent occlusion and migration are common. In a large prospective study on 222 patients with malignant biliary obstruction treated with percutaneous self-expandable metal stents, primary patency was 74.2% at 1 month, 41.9% at 3 months, and only 24.9% at 6 months; major complications, including sepsis, cholangitis and major bleeding, occurred in 10% of patients [30].

### 5.2. Conventional Endoscopy

Conventional endoscopy remains the first-line approach to biliary and pancreatic access in many patients with SAA, particularly when the papilla or bilioenteric anastomosis can still be reached. Techniques include the use of standard duodenoscopes, forward-viewing gastroscopes or pediatric colonoscopes, and more advanced platforms such as device-assisted enteroscopy (DAE). Each approach carries specific advantages and limitations, with significant variability in outcomes depending on the surgical reconstruction and local expertise.

**Standard duodenoscopes** are most effective in Billroth I reconstructions, where gastric–duodenal continuity is maintained, with ERCP success rates exceeding 90% and approaching those seen in patients with native anatomy. In contrast, in Billroth II reconstructions, duodenoscopes are less reliable because of the length and angulation of the afferent limb. Reported success rates for reaching the papilla in Billroth II reconstruction range from 70% to 91%, with overall technical success between 71% and 82% depending on the series, and the risk of adverse events such as perforation remains a concern [31]. Other recent evidence, however, indicates that the incidence of adverse events does not significantly differ between the two approaches [32].

Pros: Familiar platform; best for Billroth I and some Billroth II/GJ if papilla/anastomosis is reachable; full ERCP accessory ecosystem and elevator for cannulation.

Cons: Often cannot reach the papilla or HJ anastomosis in long-limb BII/RY reconstructions; sharp angulations risk perforation; looping; limited in Whipple/RY-HJ anatomies [33].

**Forward-viewing endoscopes** such as gastroscopes or pediatric colonoscopes have been traditionally employed in Billroth II anatomy. They permit intubation of the afferent loop with greater facility than a duodenoscope. In many series the success rate of afferent limb intubation ranges between 80 and 100%, similar to the success rate of cannulation [23]. AEs range between 5 and 10% according to the series, but perforation rate is widely below 2% [34]. Forward-viewing endoscopes, including pediatric colonoscopes and gastroscopes, have been increasingly applied not only in Billroth II reconstructions but also after pancreatoduodenectomy (Whipple). In a recent large retrospective series of 334 patients undergoing 665 ERCPs exclusively with forward-viewing instruments, procedural feasibility in Whipple anatomy was high, with successful enteroscopy in 91.5%, selective biliary cannulation in 83.6%, and therapeutic success in 81.6%. The overall adverse event rate was 5.1%, predominantly mild, further confirming the safety and reliability of forward-viewing platforms in this challenging postoperative setting [22].

Pros: Widely available; easier and safer access to the duodenal stump in BII; availability of pediatric colonoscopes with large channel, 3.8 mm.

Cons: No elevator; generally designed with smaller channels (limits stent size/devices); challenging cannulation, especially to cannulate the pancreatic duct; commonly unsuitable for long-limb RYGB/RY-HJ.

**Device-assisted enteroscopy (DAE)** represents an important innovation for performing ERCP in patients with long-limb reconstructions. Double-balloon enteroscopy (DBE) in particular provides deep intubation of the small intestine, with reported technical success rates between 70% and 90% depending on center expertise [35]. Clinical success, defined as completion of the intended therapeutic intervention, is somewhat lower, often in the range of 65–80%. Single-balloon enteroscopy (SBE) is less effective in deeply angulated anatomies but may be easier to use and more widely available [36].

A large meta-analysis including 1227 SBE–assisted ERCPs in patients with SAA (such as Roux-en-Y gastrectomy, hepaticojejunostomy, Whipple reconstruction, or Billroth II) showed encouraging results in terms of enteroscopy success (86.6%) and biliary cannulation (90%). However, when focusing on the overall procedural success, defined as the completion of the intended biliary intervention, the rate dropped to only 75.8% [37].

Pros: Designed for deep intubation of long limbs; success commonly 70–90% in expert centers; short-type scopes (155 cm) allow use of many ERCP accessories.

Cons: Time consuming, technically demanding; limited availability of enteroscopes at many centers; cannulation without an elevator is harder; accessory limitations persist (especially with longer scopes and due to the small size of the operative channel); AEs include pancreatitis, perforation, and bleeding (overall low-to-moderate, up to 15–18% in some cohorts).

### 5.3. EUS-Guided Procedures

In recent years, EUS has evolved from a purely diagnostic tool into a versatile therapeutic platform, particularly for the management of biliopancreatic diseases in patients with SAA. The growing prevalence of complex reconstructions such as Roux-en-Y gastric bypass, pancreaticoduodenectomy, and hepaticojejunostomy has underscored the limitations of conventional ERCP and device-assisted enteroscopy, fostering the development of EUS-guided alternatives. These are inherently hybrid procedures, combining real-time sonographic guidance with continuous fluoroscopic support to ensure accuracy and safety during ductal puncture, guidewire manipulation, and stent deployment. Their success requires not only advanced endoscopic expertise but also solid radiologic skills, together with the availability of a comprehensive device armamentarium, including dedicated needles, guidewires, dilation systems, and lumen-apposing metal stents, which are realistically available only in tertiary referral centers.

#### 5.3.1. EUS-Guided Rendezvous (EUS-RV)

EUS-guided rendezvous is generally employed as a rescue technique when standard cannulation fails. The procedure entails puncturing an intrahepatic bile duct under EUS guidance, advancing a guidewire across the papilla or bilioenteric anastomosis, and then retrieving it endoscopically to complete ERCP in a conventional manner. Notably, ESGE guidelines recommend the rendezvous technique over transmural or percutaneous drainage whenever anatomy is favorable, as it ensures internal biliary drainage and is associated with fewer adverse events, reinforcing its role as the preferred rescue option in this setting [38].

Its feasibility is highest in patients with Billroth II anatomy or hepaticojejunostomy, where the papilla or anastomosis is, at least, potentially reachable. The greatest technical challenge is the ability to pass the guidewire through the papilla or across a tight stricture. For this reason, long guidewires are generally recommended (better if partially or entirely hydrophilic) [38,39].

Clinical studies report technical success in approximately 80–90% of cases and clinical success in 70–85%, with AEs around 10% (cholangitis, bile leak, bleeding). When compared with PTBD, EUS-RV avoids the discomfort and morbidity of external drains; when compared with EA-ERCP, it can shorten procedure time, although it requires advanced EUS skills and remains dependent on wire passage through the stricture [40].

A recent meta-analysis on twelve studies confirms the efficacy of EUS rendezvous after failing biliary cannulation with ERCP: the pooled rate of technical success was 86.1%, the pooled rate of clinical success was 80.8%, while the pooled rate of overall adverse events was 14% [41].

In SAA, depending on the surgical reconstruction, puncture of intrahepatic bile ducts from the stomach can be performed (Figure 6). This approach ensures a good direction toward the papilla and a stable position of the echoendoscope, and is the most commonly used route [42]. Hybrid rendezvous (EUS-HRV), where a dilator is temporarily inserted to facilitate guidewire maneuverability, has been shown to increase technical success up to 90% even in difficult cases [43].

#### 5.3.2. EUS-Guided Antegrade Drainage

EUS-guided antegrade drainage is conceptually similar to interventional radiology: the technique involves puncture of a biliary duct, usually from the stomach, using a large caliber needle (usually 19- or 20-Gauge needle), guidewire passage into the biliary tree and, through the papilla/anastomosis, into the downstream bowel (Figure 7). The endoscopist can perform a dilation if necessary and remove biliary stones when present or place a self-expandable metal stent. Technical success rates above 85–90% and clinical success of around 80–90% have been reported. AEs occur in 10–15%, mostly bile leakage or cholangitis [44]. The main advantage of antegrade stenting is that it preserves a physiologic drainage route and avoids external catheters, while the drawback is that reintervention can be technically challenging once the stent is in place. Generally, there is no need to place a stent to protect the initial needle access [45].

#### 5.3.3. EUS-Guided Hepaticogastrostomy (HGS)

HGS has become a crucial therapeutic option in cases where conventional ERCP fails or is contraindicated, particularly in patients with SAA, malignant duodenal obstruction, or inaccessible ampulla. HGS creates a permanent fistula between the intrahepatic ducts (typically segment II or III) and the gastric lumen, using a hybrid SEMS specifically designed for EUS-guided HGS. It is a partially covered hybrid stent with two functional segments: the intrahepatic portion is uncovered, allowing anchorage within the liver parenchyma, while the gastric portion is fully covered and equipped with an anti-migration flange, minimizing the risk of bile leakage and stent dislodgement [38]. Before the advent of dedicated hybrid stents, and in centers where these devices are not available, plastic stents have been used (Figure 8); in this setting, a long stent is suggested to avoid dislodgement [46]. Technical aspects include careful puncture to avoid vascular structures, gentle dilation of the tract, and deployment of covered SEMS to minimize bile leak.

Systematic reviews and multicenter series report technical success above 90% and clinical success around 80–90%. AEs, occurring in 10–20%, include the risks of persistent fistula and bile leak (which may be life-threatening if not promptly recognized), bleeding and peritonitis. Compared with PTBD, HGS is associated with fewer reinterventions and greater patient comfort; compared with antegrade stenting, it is easier to repeat but involves creating a transmural fistula with its own risks. Recent meta-analyses revealed technical success around 94.4% and clinical success around 88.6%, albeit with a higher overall adverse event rate (about 23.8%). HGS can also be performed as rescue option, once an EUS-HRV or an antegrade drainage fails [47].

When EUS-HGS is combined with antegrade stenting (EUS-HGAS), the technique offers dual drainage routes (transgastric and transpapillary), potentially reducing recurrence of biliary obstruction; nonetheless, despite this theoretical advantage, success and complication rates remain similar to EUS-HGS alone (technical success near 90%, clinical success 92.5%, and AEs about 13%). In addition, EUS-HGAS is associated with significantly longer fluoroscopy time, radiation exposure, and procedure duration compared to EUS-HGS, with no clear clinical advantage [48].

The meta-analysis by Binda et al., which included 33 studies and a total of 1644 patients undergoing EUS-guided hepaticogastrostomy for malignant biliary obstruction, demonstrated high rates of efficacy and acceptable safety outcomes. The technical success rate of the procedure was 97.7%, indicating that stent placement was achieved in nearly all cases. The clinical success rate, calculated using an intention-to-treat approach, was 88.1%, reflecting effective biliary drainage and symptom relief in the majority of patients.

Regarding safety, the overall rate of procedure-related adverse events was 12.0%. The most frequently reported complications were cholangitis or sepsis, occurring in 2.8% of patients, and bleeding, observed in 2.3%. The study also highlighted that the use of dedicated stents significantly reduced the incidence of adverse events. Additionally, better technical and clinical outcomes were associated with higher procedural volume, particularly in centers performing more than four EUS-HGS procedures per year [49].

#### 5.3.4. EUS-Directed Transgastric ERCP (EDGE)

EDGE is an advanced endoscopic technique developed to overcome the limitations of ERCP in patients with RYGB, where the duodenal papilla cannot be reached by the conventional route. The concept relies on the creation of a temporary communication between the gastric pouch and the excluded stomach by the placement of a lumen-apposing metal stent (LAMS). After tract maturation and balloon dilation, a standard duodenoscope can be advanced into the remnant stomach, allowing ERCP to be performed much like in patients with normal anatomy.

In the past decade, a growing body of evidence has supported the use of EDGE. In a systematic review and meta-analysis, Dhindsa and colleagues compared EDGE, laparoscopic-assisted ERCP (LA-ERCP), and balloon-assisted enteroscopy ERCP (BE-ERCP). Their results demonstrated that EDGE and LA-ERCP both achieved excellent technical and clinical success rates (around 95%), whereas BE-ERCP showed considerably lower figures, with technical success near 70% and clinical success below 60%. This analysis emphasized how EDGE combines the high effectiveness of LA-ERCP with the minimally invasive advantages of a purely endoscopic procedure [50].

A recent meta-analysis by Gangwani et al. directly compared EDGE with LA-ERCP. While both techniques achieved similar rates of technical success and adverse events, EDGE was associated with significantly shorter procedure times, on average more than 90 min shorter, thus highlighting its procedural efficiency. These data also suggest that stent diameter is the main determinant of migration, with 15-mm stents showing a higher risk of displacement compared to 20-mm stents [51].

Based on these findings, the authors suggested that EDGE should be prioritized in expert centers where the necessary expertise and equipment are available [52].

The recent systematic review and meta-analysis by Reddy et al. evaluated the safety and efficacy of EDGE in patients with Roux-en-Y gastric bypass. Across 11 studies including 543 patients, the pooled technical success of LAMS placement was 97.8%, and ERCP completion through the stent was achieved in 95.6% of cases. Adverse events occurred in about 21.9%, with stent migration (13.3%) being the most frequent complication, followed by bleeding (6.6%), and less commonly post-ERCP pancreatitis or perforation (2.2%). Importantly, the relatively high overall AE rate was largely driven by stent migration, a complication that is often manageable endoscopically, thereby reinforcing EDGE as a highly effective and generally safe approach, although cautious follow-up remains required to manage stent-related events [53].

Overall, the available evidence indicates that EDGE is a safe, effective, and time-efficient approach for biliary and pancreatic interventions in RYGB patients. Its adoption is rapidly expanding worldwide, and in centers with sufficient expertise it is increasingly regarded as the preferred first-line option over both laparoscopic and enteroscopy-assisted alternatives.

#### 5.3.5. EUS-Directed Transenteric ERCP (EDEE)

In 2014, the first case of a trans-enteric duodeno-jejunal bypass by using a lumen-apposing metal stent (LAMS) to facilitate biliary access in Roux-en-Y hepaticojejunostomy was published [54]. To date, after more than ten years, EDEE is a consolidated and well-documented technique that represents a valid therapeutic option in patients with altered anatomy, although relegated to tertiary referral centers with experience in this type of procedure.

In this technique, a LAMS is used to create an anastomosis between the stomach or duodenum and the afferent biliary limb, thus restoring endoscopic access to the biliary anastomosis with a standard duodenoscope.

By providing a permanent and reusable access route, the entero-enteric bypass enables repeated interventions over time with standard devices. Moreover, the possibility to reach directly the anastomotic area with large-caliber scopes (e.g., pediatric colonoscope with 3.8 mm operative channel) also permits treatment of the most challenging cases with the full standard equipment (Figure 9).

The largest case series published, and subsequent expanded follow-up, report a technical success above 95% and long-term clinical efficacy above 90%. AEs included stent migration, bleeding, and leaks but were largely manageable. Compared with EA-ERCP, entero-enteric bypass is more durable and facilitates reintervention; compared with PTBD, it avoids external drains and improves quality of life [55,56].

Main limitations of EDEE are technical complexity, the need for careful preprocedural imaging to identify the correct loop, and the limited availability of centers with the required expertise. Nonetheless, it is rapidly being recognized as a paradigm-changing approach.

## 6. Management of Pancreatic Diseases in Patients with SAA

Endoscopic pancreatic interventions can be challenging in patients who have undergone pancreaticoduodenectomy (PD), Roux-en-Y gastric bypass or Roux-en-Y partial or total gastrectomy, while after Roux-en-Y hepaticojejunostomy the pancreatic duct remains easily accessible.

Following pancreatoduodenectomy, pancreatic treatment is frequently required, as complications such as pancreaticojejunostomy anastomotic strictures (PJASs), pancreatic duct stones (PDSs), and pancreatic fistulas (PFs) are not uncommon. PJAS is encountered in approximately 2% to 11% of patients undergoing PD [57], while clinically relevant leaks of jejunal-pancreatic anastomosis occur in 17–18% of cases [58,59].

In those cases, access to the pancreaticojejunostomy (PJ) is technically demanding because this anastomosis is located deep in the afferent limb beyond the hepaticojejunostomy anastomosis, which makes endoscope insertion to the anastomosis site harder [60]. Moreover, pancreatic duct access is often complicated by complete obstruction or dehiscence of PJ anastomosis.

For pancreatic leaks, the pancreatic fluid outflow can be internalized through EUS-guided transmural drainage of the peripancreatic fluid collection; several studies [61] have reported clinical success rates of approximately 90–100%, with better outcomes than percutaneous approach in terms of fistula resolution, time to clinical success, adverse events, postoperative length of stay, and prevention of recurrence [61,62].

Data on the most suitable timing of intervention for post-operative pancreatic fluid collections (POPFCs) are still relatively scant. Two retrospective studies have assessed the impact of different timing strategies for endoscopic ultrasound EUS-guided drainage of POPFCs. The first compared three different timings: acute drainage (<2 weeks), early drainage (<4 weeks) and delayed drainage (>4 weeks) [63]; the latter showed results of early drainage (<4 weeks) for POPFCs, with a subgroup analysis between acute (<2 weeks) versus early drainage (2–4 weeks) [64]. Both studies found no differences in terms of clinical success and AEs among these different timings. A more recent meta-analysis confirmed absence of significant differences in efficacy and safety [65]. However, early and acute drainage should be considered in selected cases, such as severe sepsis.

For pancreaticojejunostomy strictures, EUS-guided pancreatic duct drainage has nowadays become a key intervention [66,67]. As described above for the biliary tract, EUS-guided pancreatic drainage includes two main techniques: EUS-transmural duct drainage (EUS-TMD) and EUS-RV [67].

EUS-TMD comprises two strategies: EUS-guided pancreaticogastrostomy, in which a stent is placed between the pancreatic duct and the gastrointestinal lumen, and transenteric antegrade stenting, which involves advancing the stent further into the small intestine through the anastomotic site. Initially described as a rescue option following failed ERCP, EUS-guided interventions have subsequently been reported as a viable first-line therapy [68,69,70]. A retrospective single-arm observational study by Oh and colleagues evaluated the long-term outcomes of EUS-PD in patients with PJAS after Whipple surgery in whom endoscopic retrograde pancreatography (ERP) had failed. Direct EUS-guided transanastomotic placement of a plastic stent was effective only in 3/23 (13%), but in all these cases, transmural FCSEMS placement was successful in all patients. In 11/20 (30%) the transmural FCSEMS was left permanently in place, while in 9/20 cases it served as a bridge therapy for a subsequent transanastomotic plastic stent placement [71].

A retrospective study compared the outcomes of ERP and EUS-guided PD drainage in patients with pancreaticojejunostomy stricture. EUS-guided PD drainage was performed with either transmural stent insertion (52.5%) or antegrade stenting (40%), while EUS-guided rendezvous was performed in 7.5% of patients. This study found a significant advantage of EUS-guided approach over ERP [69].

A more recent systematic review by Basiliya and colleagues confirmed these findings. EUS-guided approach was superior to ERP-guided approach in terms of pancreatic duct opacification (87% vs. 30%; *p* < 0.001), cannulation success (79% vs. 26%; *p* < 0.001), and stent placement (72% vs. 20%; *p* < 0.001) [72].

EUS-RV has also been described, and it can be an option [73]. In our view, this approach is especially indicated for pancreatic fistulas when more complex retrograde treatments are needed, for example, in case of necrosis of the jejunal stump [74].

## 7. Conclusions

Patients with surgically altered anatomy represent one of the most complex and heterogeneous populations encountered in biliopancreatic endoscopy. Conventional ERCP, although still the first-line approach in selected reconstructions, is frequently limited by anatomical barriers, long afferent limbs, and unfavorable angulations, resulting in high technical failure and complication rates. Device-assisted enteroscopy has expanded the therapeutic options, but its procedural success remains modest, particularly in long-limb reconstructions and complex post-surgical strictures.

In this scenario, EUS has progressively evolved from a purely diagnostic tool into a central therapeutic platform. EUS-guided rendezvous, antegrade drainage, hepaticogastrostomy, and combined approaches have shown high rates of technical and clinical success, often comparable or superior to interventional radiology, with the added advantage of avoiding external drains and improving quality of life. Novel procedures such as EDGE and EDEE have further revolutionized the management of Roux-en-Y gastric bypass and complex bilioenteric anastomoses, offering durable and reusable access to the biliary system.

Despite these advances, EUS-guided interventions remain technically demanding, requiring not only advanced endoscopic skills but also solid radiologic expertise and a full spectrum of dedicated devices. Outcomes are strongly influenced by operator experience and institutional volume, underlining the importance of centralization in high-volume referral centers.

Future perspectives will likely include further refinement of dedicated stents, accessories, and endoscopes designed for altered anatomy, together with the implementation of structured training programs and the development of standardized, evidence-based guidelines to enhance safety, reproducibility, and overall outcomes. Multidisciplinary collaboration among endoscopists, radiologists, and surgeons remains pivotal to ensure optimal patient selection and to manage adverse events.

In conclusion, EUS has definitively changed the treatment paradigm of biliopancreatic diseases in surgically altered anatomy. While conventional endoscopic and radiologic techniques maintain a role, EUS-guided procedures increasingly represent the preferred minimally invasive approach, provided that expertise and resources are available. Ongoing research, training, and technological development will be essential to consolidate its role as the standard of care in this challenging patient population.

## Figures and Tables

**Figure 1 diagnostics-15-02707-f001:**
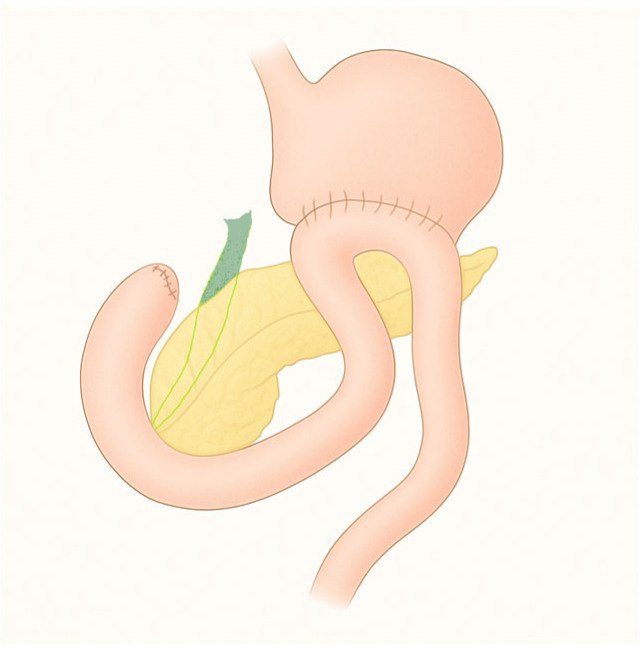
Billroth II gastrectomy.

**Figure 2 diagnostics-15-02707-f002:**
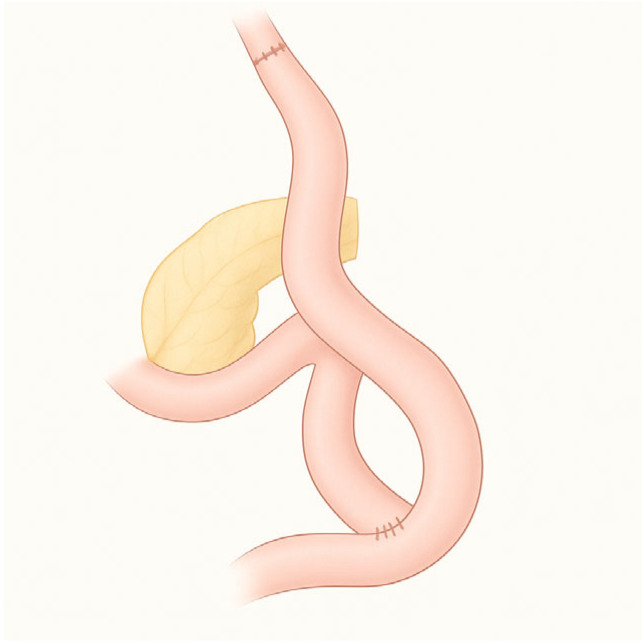
Roux-en-Y total gastrectomy.

**Figure 3 diagnostics-15-02707-f003:**
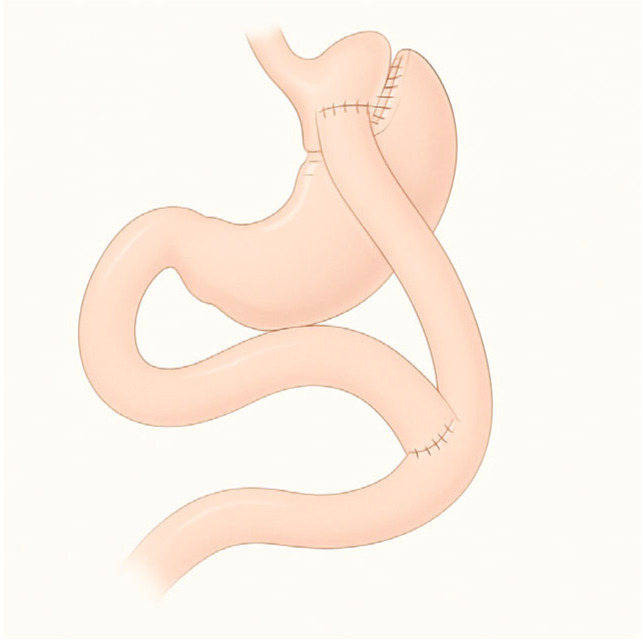
Gastric bypass.

**Figure 4 diagnostics-15-02707-f004:**
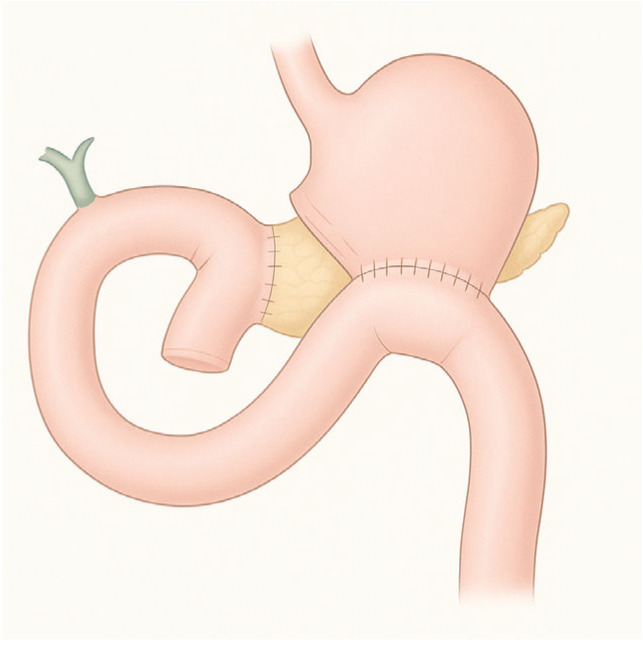
Pancreaticoduodenectomy.

**Figure 5 diagnostics-15-02707-f005:**
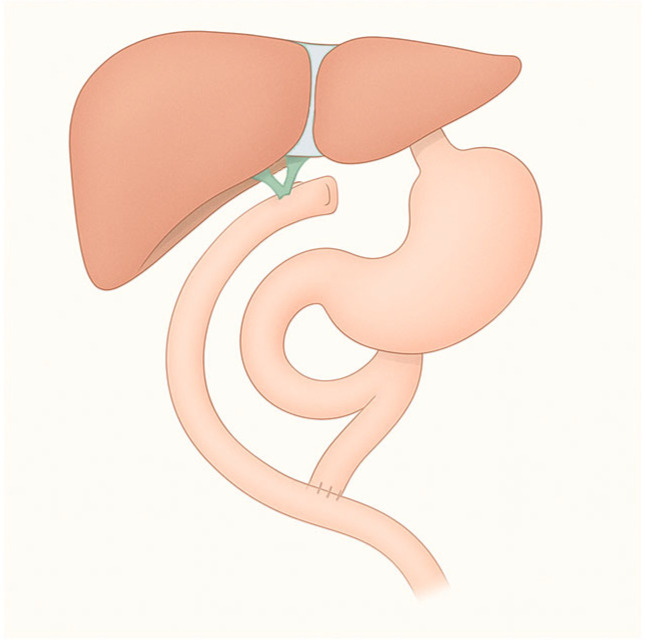
Hepaticojejunostomy.

**Figure 6 diagnostics-15-02707-f006:**
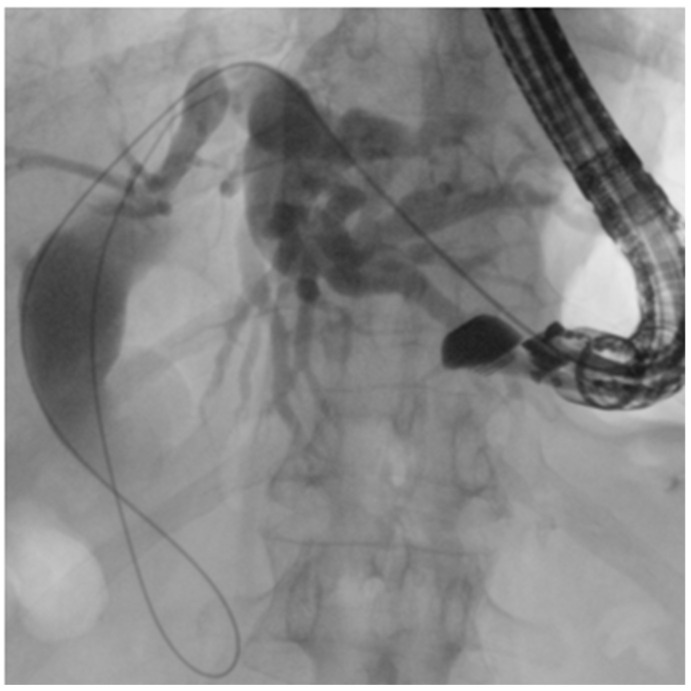
EUS-guided rendezvous: puncture of the intrahepatic bile ducts (IHBD) from the stomach.

**Figure 7 diagnostics-15-02707-f007:**
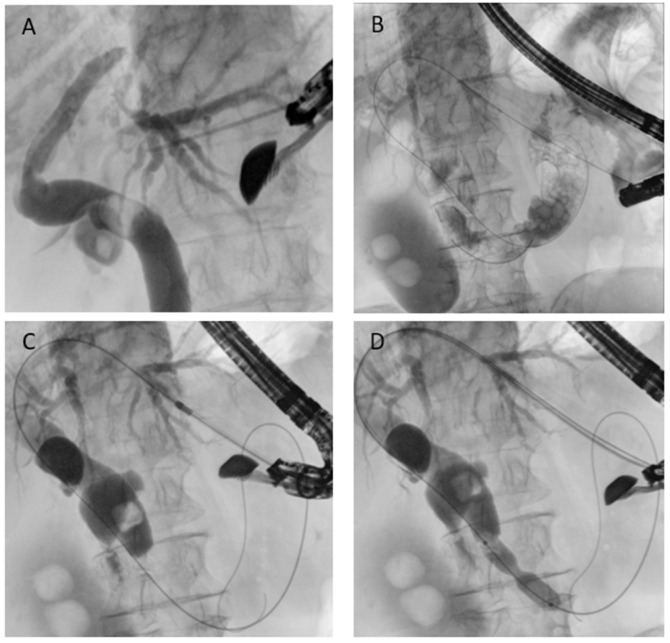
EUS-guided antegrade drainage. (**A**) Puncture of the intrahepatic duct; (**B**) guidewire passage into the biliary tree and across the papilla; (**C**) dilation of the gastric access using a cystotome; (**D**) endoscopic papillary dilation performed with a biliary balloon dilatation catheter.

**Figure 8 diagnostics-15-02707-f008:**
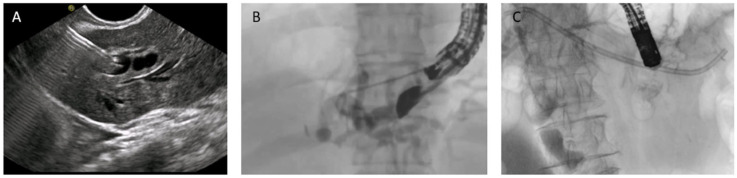
EUS-guided hepaticogastrostomy (HGS). (**A**) Puncture of the intrahepatic duct, EUS view; (**B**) puncture of the intrahepatic duct, radiologic image; (**C**) placement of a plastic stent to ensure biliary drainage.

**Figure 9 diagnostics-15-02707-f009:**
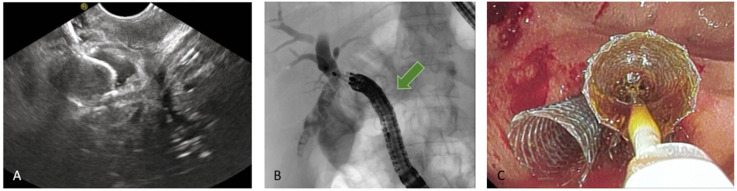
EUS-guided duodenojejunal anastomosis (EDEE): (**A**) EUS-guided deployment of a LAMS into the jejunal lumen; (**B**) use of a pediatric colonoscope to reach the biliojejunal anastomosis through the LAMS (indicated by the green arrow); (**C**) fcSEMS protruding from the anastomosis.

## Data Availability

No new data were created or analyzed in this study. Data sharing is not applicable to this article.

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
