# Peer review of "Role of the EUS in the Treatment of Biliopancreatic Disease in Patients with Surgically Altered Anatomy"

_diagnostics, 2025, doi:10.3390/diagnostics15212707_

Round 1

Reviewer 1 Report

Comments and Suggestions for Authors

This timely review details EUS's shift from a diagnostic to a primary therapeutic tool for biliopancreatic diseases in surgically altered anatomy. It highlights advanced techniques like EDGE and EDEE, supported by meta-analyses showing superiority over alternatives, and appropriately stresses centralizing these complex procedures in high-volume centers..

Areas for Fine-Tuning:
While the literature search is commendably up-to-date, integrating specific recommendations from the latest international guidelines (e.g., European Society of Gastrointestinal Endoscopy (ESGE)) would further solidify the authors' conclusions. The methods section could be more transparent by detailing the search strategy. Minor inconsistencies in terminology (e.g., SAA vs. spelled out) and figure referencing are present but easily correctable.

Overall, this is a high-quality review that accurately reflects the state of the art. The suggestions below are aimed at polishing it into an definitive reference in the field.

Tracked-Changes Style Critique (Targeted to Manuscript Sections)

Section 2: Materials and Methods

  • Comment:The search strategy is described but lacks specific detail, making it difficult to assess reproducibility. A more systematic approach description is recommended for review articles.

Suggestion: Specify the exact Boolean operators used (e.g., ("endoscopic ultrasound" OR EUS) AND ("altered anatomy" OR "gastric bypass") AND ("drainage" OR "EDGE" OR "rendezvous")). Mentioning a specific review and reporting framework like PRISMA-ScR (for Scoping Reviews) (Page MJ, et al. The PRISMA 2020 statement: an updated guideline for reporting systematic reviews. BMJ. 2021;372:n71. doi:10.1136/bmj.n71) would enhance methodological quality.

Section 3: Postsurgical Anatomy

  • It would be useful to have a figure, an explanatory drawing of the types of Postsurgical Anatomy in this section; so that even the less experienced clinicians can have a precise idea of ​​the technical difficulty in performing these procedures and can explain it to patients.

Section 4: Diagnostic EUS and EUS-TA in Altered Anatomy

  • Comment:This section accurately describes the limitations of diagnostic EUS in SAA. A recent large systematic review and meta-analysis specifically addresses this challenge and provides pooled success rates, which could be cited to generalize the findings beyond the individual studies mentioned (Brozzi et al., Wilson et al.).

Suggestion: Incorporate data from the recent meta-analysis by Tanisaka et al. (2021; DOI: 10.3390/jcm10081624) and Dell’Anna (2025, DOI:  10.1177/17562848251359006) to provide a broader, more quantitative context for the technical challenges and success rates of EUS-TA in SAA.

Section 5: Therapeutic Biliary Endoscopy in SAA & Section 7: Conventional Endoscopy

  • Comment:The failure rates cited for ERCP in various anatomies are well-supported. The recent large multicenter study by Han et al. (2025) cited here is an excellent source. The pros/cons tables for different scopes are very useful.
  • Suggestion:The section on DAE could briefly mention the latest developments with motorized spiral enteroscopy (MSE), noting both its potential for faster insertion and the recent safety concerns highlighted in the cited reference (Moreels et al., 2024). This presents a balanced view.
  • Reference:(Already well-cited with Han et al. and Moreels et al.)

Section 8.1: EUS-Guided Rendezvous (EUS-RV)

  • Comment:The technical success rate of 86.1% from the cited meta-analysis (Klair et al., 2023) is appropriate. It's worth noting that international guidelines now often position EUS-RV as a primary salvage technique over percutaneous approaches in expert hands.
  • Suggestion:Consider referencing a guideline that formalizes this preference, emphasizing the advantage of internal drainage.

Section 8.3: EUS-Guided Hepaticogastrostomy (HGS)

  • Comment: The citation of the large 2024 meta-analysis by Binda et al. (DOI: 10.1055/A-2282-3350) is excellent and provides the most current and robust evidence on EUS-HGS outcomes. This is a major strength of the manuscript.
  • Suggestion:The text could more explicitly state that the use of dedicated, electrocautery-enhanced LAMS is now considered a best practice and is a key factor in the high technical success and improved safety profile, as highlighted in the Binda analysis.

Section 8.4: EUS-Directed Transgastric ERCP (EDGE)

  • Comment:This section is very well-written and supported by excellent, recent meta-analyses (Dhindsa 2020, Gangwani 2024, Reddy 2024). The conclusion that EDGE is a first-line option in expert centers is fully justified by this evidence.
  • Suggestion:A minor point: When discussing the Reddy et al. meta-analysis, it might be helpful to specify that the 21.9% AE rate is heavily weighted by stent migration, which is often a manageable complication, thus reinforcing the procedure's overall safety.
  • Reference:(Perfectly cited with the most recent reviews).

Section 9: Pancreatic Diseases

  • Comment:This is a complex and less standardized area. The paper does a good job covering the options. The reference to the systematic review by Basiliya et al. (2021) is crucial as it provides strong evidence for the superiority of EUS-guided over ERP-guided approaches for pancreaticojejunostomy stenosis.
  • Suggestion:The sentence on EUS-RV for pancreatic fistulas is a bit unclear ("...in case of necrosis of the jejunal attempted if a retrograde method is needed"). This should be rephrased for clarity.
  • Suggestion:Consider also citing a more recent review or large series that consolidates the role of EUS-guided drainage for postoperative pancreatic collections and fistulas.
  • Reference:Basiliya K, et al. Endoscopic retrograde pancreatography-guided versus endoscopic ultrasound-guided technique for pancreatic duct cannulation in patients with pancreaticojejunostomy stenosis: a systematic literature review. Endoscopy. 2021;53(3):266-276. doi:1055/a-1200-0199.
  • Additional Reference: 
  • 1.         Chandan, Endosc Int Open 2020, 8, E1664-E1672, doi:10.1055/a-1236-3350.
  • 2.         Sundaram, S. World journal of gastrointestinal endoscopy 2023, 15, 122-132, doi:10.4253/wjge.v15.i3.122.
  • 3.         Will, U.. J Clin Med 2024, 13, doi:10.3390/jcm13247709.
  • 4.         Wang, R.. Surgical laparoscopy, endoscopy & percutaneous techniques 2025, 35, doi:10.1097/SLE.0000000000001359.
  •  
  • Figures & Terminology
  • Comment:The figures are described well. A minor suggestion would be to ensure all figure parts (A, B, C, D) are explicitly mentioned and described in the caption text for maximum clarity.
  • Comment:The abbreviation "SAA" is used most of the time, but the full term "surgically altered anatomy" is sometimes used interchangeably. For consistency and readability, using the abbreviation "SAA" after its first definition is recommended.
  • Suggestion:Perform a quick check to standardize the use of "SAA" throughout the manuscript.
  • TYPOS: RAW: 124, 257

Reviewer 2 Report

Comments and Suggestions for Authors

The manuscript is a very extensive review of the role of interventional endoscopy ultrasound for drainage of obstructed bile ducts in altered surgical anatomy. Although the caseload is much smaller, it can also be used for drainage of the obstructed pancreatic duct and for drainage of ntraabdominal fluid collections.

In my opinion, the article is excessively extensive. Access to the bile duct in Billroth I and II gastrectomies can be performed conventionally. In fact, there is extensive scientific literature regarding Billroth II since almost the beginning of therapeutic ERCP in the 1980s, and the role of interventional endoscopy ultrasound is relatively small and similar to when it is used after failure of conventional ERCP. Consider removing or reducing this section.

Regarding other major disruptions of the intestinal continuity, such as Roux-en-Y, the manuscript describes interventional endoscopy ultrasound techniques very well. Perhaps an explanatory drawing of the digestive tract's appearance after these surgeries would be helpful.

Beginning on line 302, the use of the spiral enteroscope is described. Since this instrument has been withdrawn from the market, this entire explanation should be omitted.

Figure 1B discusses bile duct puncture from the duodenal bulb. Typically, the duodenal bulb is absent in the altered anatomy, so this can be confusing.

Finally, there are many paragraphs with spelling errors consisting of words joined together, for example, on lines 257, 134, 120, 112, 66, 63, 60.

On line 72, the correct spelling should be “pancreatectomy.”

Comments on the Quality of English Language

there are many paragraphs with spelling errors consisting of words joined together, for example, on lines 257, 134, 120, 112, 66, 63, 60.
